# Role of TGF-β and p38 MAPK in TSG-6 Expression in Adipose Tissue-Derived Stem Cells In Vitro and In Vivo

**DOI:** 10.3390/ijms25010477

**Published:** 2023-12-29

**Authors:** Hye Youn Kwon, Yongdae Yoon, Ju-Eun Hong, Ki-Jong Rhee, Joon Hyung Sohn, Pil Young Jung, Moon Young Kim, Soon Koo Baik, Hoon Ryu, Young Woo Eom

**Affiliations:** 1Department of Surgery, Yonsei University Wonju College of Medicine, Wonju 26426, Republic of Korea; kwonhy@yonsei.ac.kr (H.Y.K.); surgery4trauma@yonsei.ac.kr (P.Y.J.); 2Regeneration Medicine Research Center, Yonsei University Wonju College of Medicine, Wonju 26426, Republic of Korea; yongdae0611@naver.com (Y.Y.); drkimmy@yonsei.ac.kr (M.Y.K.); baiksk@yonsei.ac.kr (S.K.B.); 3Department of Biomedical Laboratory Science, College of Software and Digital Healthcare Convergence, Yonsei University Mirae Campus, Wonju 26493, Republic of Korea; jehong@yonsei.ac.kr (J.-E.H.); kjrhee@yonsei.ac.kr (K.-J.R.); 4Department of Convergence Medicine, Yonsei University Wonju College of Medicine, Wonju 26426, Republic of Korea; sjh@yonsei.ac.kr; 5Department of Internal Medicine, Yonsei University Wonju College of Medicine, Wonju 26426, Republic of Korea

**Keywords:** mesenchymal stem cells, TSG-6, TGF-β, inflammation, macrophages

## Abstract

Mesenchymal stem cells (MSCs) regulate immune cell activity by expressing tumor necrosis factor-α (TNF-α)-stimulated gene 6 (TSG-6) in inflammatory environments; however, whether anti-inflammatory responses affect TSG-6 expression in MSCs is not well understood. Therefore, we investigated whether transforming growth factor-β (TGF-β) regulates TSG-6 expression in adipose tissue-derived stem cells (ASCs) and whether effective immunosuppression can be achieved using ASCs and TGF-β signaling inhibitor A83-01. TGF-β significantly decreased TSG-6 expression in ASCs, but A83-01 and the p38 inhibitor SB202190 significantly increased it. However, in septic C57BL/6 mice, A83-01 further reduced the survival rate of the lipopolysaccharide (LPS)-treated group and ASC transplantation did not improve the severity induced by LPS. ASC transplantation alleviated the severity of sepsis induced by LPS+A83-01. In co-culture of macrophages and ASCs, A83-01 decreased TSG-6 expression whereas A83-01 and SB202190 reduced Cox-2 and IDO-2 expression in ASCs. These results suggest that TSG-6 expression in ASCs can be regulated by high concentrations of pro-inflammatory cytokines in vitro and in vivo, and that A83-01 and SB202190 can reduce the expression of immunomodulators in ASCs. Therefore, our data suggest that co-treatment of ASCs with TGF-β or p38 inhibitors is not adequate to modulate inflammation.

## 1. Introduction

Mesenchymal stem cells (MSCs) are important cells that produce tumor necrosis factor-α (TNF)-α stimulated gene-6 (TSG-6), which has been identified as a biomarker that can be used to predict the efficacy of human MSCs as inflammation modulators [1]. When an injury occurs, the main therapeutic mechanism of action of MSCs is through paracrine signals, which results in the promotion of angiogenesis, prevention of apoptosis, suppression of the immune response, and regulation of fibrosis at the site of injury [2,3,4]. Among these effects, the immunomodulatory effects of MSCs are the focus of treatment strategies in regenerative medicine, including inflammatory diseases [5,6,7]. MSCs have the ability to migrate to the site of injury, but only 0.1–1% of transplanted MSCs have been found at the target site [8,9,10]. However, even if MSCs do not migrate to the site of injury, they can be exposed to inflammatory cytokines and secrete various paracrine factors, such as indoleamine 2,3-dioxygenase (IDO), prostaglandin E_2_, TSG-6, nitric oxide (NO), interleukin (IL)-6, IL-10, and IL-1 receptor antagonist (IL-1ra), to modulate the activities of various immune cells [11,12]. MSCs can be isolated and cultured from adult tissues, such as bone marrow, peripheral blood, and adipose tissue, and prenatal tissues, such as the umbilical cord and placenta [13,14,15,16], but each MSC type shows slight differences in cell biological characteristics, proliferation ability, and surface marker expression [17,18,19]. Regarding the immunomodulation and anti-inflammatory effects of MSCs, bone marrow-derived MSCs (BMSCs) show a higher therapeutic effect than adipose tissue-derived stem cells (ASCs) on airway inflammation [20] and endotoxic shock [21]. However, in atherosclerosis, ASCs have been reported to have better anti-inflammatory effects than BMSCs [22]. Additionally, it has been reported that ASCs can inhibit the proliferation of peripheral blood mononuclear cells and the differentiation of monocytes into dendritic cells more strongly than BMSCs [23,24]. Moreover, considering that adipose tissue is an easily accessible, abundant, and reproducible source of MSCs, ASCs have greater appeal in clinical research than BMSCs [25,26].

TSG-6 is a protein that exhibits tissue-protective and anti-inflammatory properties [27] and is expressed by cells including monocytes, neutrophils, macrophages, dendritic cells, and MSCs [1,28,29,30]. TSG-6 interacts with various extracellular matrix (ECM) proteins, including fibronectin, to stabilize and remodel the ECM [31,32] and can alleviate fibrosis by downregulating the phosphorylation of suppressor of mothers against decapentaplegic homolog (SMAD) [30,33]. In addition, TSG-6 plays an anti-inflammatory role by regulating the migration and proliferation of endothelial cells, neutrophils, mast cells, macrophage foam cells, vascular smooth muscle cells, and macrophages [34,35]. Its anti-inflammatory effects have been demonstrated in animal models of myocardial infarction and liver injury [1,29,36,37,38,39,40]. The anti-inflammatory mechanisms of TSG-6 on macrophages include regulating migration and proliferation [34], decreasing nuclear factor-κB (NF-κB) translocation [41], increasing prostaglandin production [42], modulating the activity of p38 and c-Jun N-terminal kinase (JNK) [34], and inducing M1-to-M2 macrophage transition [43,44].

In 2009, it was reported that intravenously injected MSCs could improve myocardial infarction by expressing TSG-6 even when embolized in the lung [37]. Since then, many studies have shown the immunomodulatory and reparative effects of TSG-6 in various diseases, such as atherosclerosis, myocardial infarction, corneal wounding, colitis, acute or chronic liver injury, acute lung injury, and rheumatoid arthritis [1,29,37,38,40,45,46,47]. Low concentrations (1 ng/mL) of TNF-α and IL-1β alone have been shown to slightly increase TSG-6 expression, but STAT-activating or interferon-γ (IFN-γ) + lipopolysaccharide (LPS) treatment conditions significantly increased TSG-6 expression in ASCs [30]. TSG-6 expression is also significantly increased in ASCs co-cultured with M1 macrophages [30]. These results suggest that IFN-γ + LPS-differentiated M1 macrophages increase the expression of TSG-6 in ASCs by increasing expression of TNF-α and IL-1β. However, once an inflammatory response is initiated in the animal body, the expression of pro-inflammatory cytokines, including that of TNF-α and IL-1β, increases but is accompanied by a compensatory induction of anti-inflammatory cytokines such as transforming growth factor-β (TGF-β) [48,49]. Therefore, MSC therapy is usually performed under mixed conditions of pro- and anti-inflammatory cytokines.

Although it is well known that MSCs exposed to pro-inflammatory cytokines in an inflammatory environment exhibit immunomodulation, the effect of anti-inflammatory cytokines on the therapeutic effects of MSCs has been underexplored. TNF-α can increase the expression of TGF-β and vice versa [50,51,52]. TNF-α and TGF-β cooperate during epithelial–mesenchymal transition [53,54] and have antagonistic effects on ECM remodeling [54,55,56]. Therefore, we investigated whether TGF-β regulates the increased expression of TSG-6 in ASCs using a combination of pro-inflammatory cocktails (IFN-γ, LPS, TNF-α, and IL-1β) in vitro and examined whether inhibition of TGF-β signaling alters the survival rate in an animal model of sepsis in vivo. In this study, we aimed to provide optimal conditions for TSG-6 expression in ASCs and evaluate the feasibility of combining TGF-β signaling inhibitors with ASCs in clinical practice. We hope to propose strategies to increase the therapeutic efficacy of ASCs.

## 2. Results

### 2.1. Anti-Inflammatory Effects of TSG-6 in M1 Macrophages

In macrophages, TSG-6 reduces the activity of NF-κB [41], which is known to induce the expression of the pro-inflammatory cytokine IL-1β [57]. Inflammatory M1-macrophages were treated with TSG-6 (0–80 ng/mL) to investigate the change in expression of the active form of cleaved IL-1β (cIL-1β). TSG-6 above 20 ng/mL significantly reduced the expression of cIL-1β (Figure 1A), and TSG-6 at 80 ng/mL significantly reduced the expression of *IL-1β* mRNA (Figure 1B). As TSG-6 can induce the M1-M2 transition of macrophages [54,56], we analyzed changes in M1 and M2 macrophage populations with flow cytometry. The population positive for CD68, an M1 marker, decreased by approximately 9.7%, but the population positive for CD206, an M2 marker, increased by approximately 17% with TSG-6 (Figure 1C). These results suggest that TSG-6 can alleviate inflammation by inhibiting the expression of pro-inflammatory cytokines, including IL-1β, and inducing the M1-to-M2 transition in macrophages.

### 2.2. Suppression of TSG-6 Expression by TGF-β in ASCs

TNF-α and TGF-β have opposing roles in ECM remodeling and inflammatory responses. TNF-α has pro-inflammatory effects and increases TSG-6 expression in MSCs [30]. Therefore, we analyzed the effect of TGF-β on the regulation of TSG-6 expression in ASCs. ASCs showed a markedly increased expression of TSG-6 when co-cultured with M1 macrophages or exposed to the four pro-inflammatory factors (4F) IFN-γ, LPS, TNF-α, or IL-1β (Figure 2A). However, TGF-β at concentrations of 0.4–2.5 ng/mL induced a decrease in TSG-6 expression with 4F in ASCs (Figure 2B). Furthermore, increasing the concentrations of TNF-α and IL-1β in 4F by 1–10 ng/mL resulted in a concentration-dependent increase in TSG-6 expression in ASCs. Co-treatment with 1 or 10 ng/mL of TGF-β revealed that TGF-β at 10 ng/mL slightly reduced the expression of TSG-6 compared with that at 1 ng/mL, though this difference was not significant (Figure 2C). These results suggest that in an inflammatory environment, inflammatory cytokines increase TSG-6 expression in a concentration-dependent manner, but TGF-β induces a decrease in TSG-6 expression in ASCs in a less concentration-dependent manner.

### 2.3. Modulation of TSG-6 Expression by TGF-β Signaling Inhibitors in ASCs

MSCs express TGF-β to induce M1-to-M2 transition of macrophages and reduce inflammation during sepsis [58]. The TGF-β signaling pathway is mediated by SMAD, JNK, ERK, ROCK, NF-κB, GSK-3β, and/or AKT [59,60]. To investigate the molecules that regulate TSG-6 expression in ASCs, ASCs were treated with TGF-β signaling inhibitors along with 4F, and the changes in TSG-6 expression were analyzed in the presence or absence of exogenous TGF-β. In ASCs, *TGF-β* mRNA was decreased by 4F, but its expression was increased when co-cultured with macrophages (Figure 3A). In the absence of exogenous TGF-β, the expression of TSG-6 in 4F-treated ASCs was reduced by inhibitors of JNK, ERK, ROCK, NF-κB, GSK-3β, and AKT (SP600125, PD98059, Y-27632, Bay 11-7082, CHIR99021, and LY294002, respectively). It should be noted that the TGF-β type I receptor inhibitor A83-01 and the p38 inhibitor SB202190 significantly increased the expression of TSG-6 (Figure 3B). Furthermore, even in the presence of exogenous TGF-β, A83-01 and SB202190 significantly increased TSG-6 expression in ASCs (Figure 3C). In ASCs, 4F and TGF-β induced the phosphorylation of p38 and SMAD2, respectively, and the phosphorylation of p38 by 4F was reduced by A83-01 (Figure 3D). These results suggest that inflammatory cytokines increase the expression of TSG-6 in ASCs by regulating the activity of p38, and although TGF-β did not induce p38 phosphorylation in ASCs, A83-01 could increase the expression of TSG-6 by decreasing 4F-induced p38 phosphorylation. In other words, TGF-β and p38 MAPK are likely to regulate TSG-6 expression in ASCs independently. Furthermore, these results suggest that use of the combination of A83-01 and SB202190 when transplanting ASCs into inflammatory animal models may increase TSG-6 expression in ASCs, leading to more effective immunomodulation.

### 2.4. Effects of A83-01 or ASCs in LPS-Induced Septic Mice

Since A83-01 significantly increased the expression of TSG-6 by reducing the phosphorylation of SMAD2 and p38 in ASCs (Figure 3) and TSG-6 can suppress inflammation, we investigated whether treatment of A83-01 and ASCs could effectively control inflammation through TSG-6 overexpression in LPS-induced septic mice. LPS-induced septic mice were treated with A83-01, ASCs, or A83-01 + ASCs and the changes in survival were analyzed. Survival of the LPS-treated group was 60% on day 1, 47% on day 2, and 0% on day 3. Contrary to expectations, transplantation of ASCs did not alter survival with LPS treatment. However, A83-01 treatment led to a faster increase in LPS-induced mortality, and ASC transplantation significantly reduced LPS + A83-01-induced mortality (Figure 4). These results suggest that the use of A83-01 in inflammatory diseases should be carefully considered as it may further increase the severity of inflammation, and the immunomodulatory function of ASCs may not be present in certain inflammatory settings.

### 2.5. Effects of A83-01 or SB202190 on ASCs Co-Cultured with Macrophages

ASCs co-cultured with macrophages or exposed to 4F and A-83-01 significantly increased TSG-6 expression; however, the survival rate of ASC- or ASCs + A83-01-treated groups of septic mice did not increase over the survival rate with LPS alone. In particular, to explain the failure of ASCs + A83-01 to increase the survival of septic mice, we examined changes in the expression of the immunomodulators TSG-6, Cox-2, and IDO2 in ASCs co-cultured with macrophages. Unlike in single-cultured ASCs, TGF-β did not induce a decrease in TSG-6 expression in ASCs co-cultured with M1 macrophages (Figure 5 Lane 3). A83-01 reduced the expression of Cox-2 in the presence of exogenous TGF-β (Figure 5 Lane 6). Moreover, SB202190 increased TSG-6 expression but decreased the expression of Cox-2 in the presence or absence of exogenous TGF-β (Figure 5 Lane 7). These results suggest that in the early inflammatory environment in vivo, TSG-6 expression in ASCs may be controlled by a high concentration of pro-inflammatory cytokines rather than a low TGF-β concentration. Furthermore, we investigated whether the levels of IL-10, IL-6, or prostaglandin E2 (PGE2), which exert anti-inflammatory effects in addition to TGF-β, could affect the expression of TSG-6 in ASCs. Macrophages secrete TGF-β and IL-10 [61,62], whereas MSCs produce TGF-β, IL-6, and PGE2, which may play an anti-inflammatory role [63,64,65]. As expected, PGE2, whose production was increased by Cox-2 activity, enhanced the expression of TSG-6 in ASCs treated with 4F, whereas IL-6 and IL-10 did not affect the expression of TSG-6 in ASCs (Figure 6A). Increased Cox-2 activity leads to enhanced PGE2 production, which can further promote TSG-6 expression in ASCs. We next investigated whether the inhibition of Cox-2 and IDO regulates the expression of TSG-6 in 4F-treated ASCs. PGE2 increased the expression of TSG-6 in 4F-treated ASCs but the Cox-2 inhibitor celecoxib markedly decreased the expression of TSG-6. The IDO inhibitor 1-methyl tryptophan increased TSG-6 expression in 4F-treated ASCs (Figure 6B). Increased Cox-2 activity increases TSG-6 expression in ASCs but the activity of IDO2 may play a role in decreasing the expression of TSG-6 in ASCs. Taken together, the therapeutic effects of ASCs may be regulated not only by TSG-6 but also by COX-2 and IDO2, suggesting that a comprehensive analysis of immunomodulators is necessary when co-treating ASCs with chemical inhibitors. To summarize, these results suggest that co-administration of A83-01 or SB202190 during ASC transplantation may be undesirable because it may decrease the expression of Cox-2 and IDO2 in ASCs, thereby reducing the therapeutic effects of ASCs.

## 3. Discussion

In this study, we observed that TGF-β decreased TSG-6 expression in ASCs through inflammatory factors, including TNF-α, and that the TGF-β type I receptor inhibitor A83-01 and p38 MAPK inhibitor SB202190 could significantly increase TSG-6 expression in ASCs. However, A83-01 treatment in LPS-induced septic mice further exacerbated sepsis. ASCs significantly increased the survival rate of LPS-induced septic mice whose sepsis was exacerbated by A83-01, but did not increase it beyond that of LPS-induced septic mice. These results can be attributed to A83-01 and SB202190 inducing a reduction in the expression of Cox-2 and IDO2 in ASCs co-cultured with macrophages. Therefore, when ASCs are used to treat inflammatory diseases, co-administration of A83-01 or SB202190, which may affect the inflammatory response, is not recommended.

It has been suggested that blood levels of TGF-β in patients with non-alcoholic fatty liver disease may be an effective biomarker of coronavirus disease 2019 (COVID-19) severity and adverse outcomes [66]. It has also been suggested that TGF-β blockade should be used to treat COVID-19 [67]. COVID-19 induces a cytokine storm, leading to acute respiratory distress syndrome [68]. MSCs secrete TGF-β, which in turn, regulates the proliferation, differentiation, migration ability, and immunomodulatory properties of MSCs [63,69]. In particular, MSC-derived TGF-β can modulate the activity of T cells, NK cells, mast cells, and macrophages/microglia [70,71,72,73,74]. Therefore, these findings of the previous reports and those of our study that stem cells overexpress TSG-6, which has anti-inflammatory functions when exposed to inflammatory cytokines, in combination with TGF-β type I receptor inhibitor A83-01, may be used to develop a therapy to alleviate the cytokine storm.

However, A83-01 exacerbated LPS-induced sepsis and ASCs were completely ineffective in ameliorating LPS-induced sepsis. The combination of A83-01 and ASCs significantly alleviated A83-01 + LPS-induced sepsis, but A83-01 exacerbated sepsis and ASCs were unable to improve LPS-induced sepsis when applied individually, suggesting that it is not advisable to combine A83-01 and ASCs to alleviate the cytokine storm. In addition, although we were unable to analyze the molecular mechanisms of co-administration of A83-01 and ASCs because of the death of the animals, we found that exposure to A83-01 or SB202190 in co-cultures of ASCs and macrophages reduced the expression of Cox-2 and IDO2 in ASCs. As Cox-2 and IDO2 are expressed by ASCs and may be important in inducing immunomodulation, it is possible that in septic mice, A83-01 was unable to alleviate the inflammatory response by inducing a reduction in the expression of TSG-6, Cox-2, and IDO2 in ASCs. The cytokine storm was expected to be more severe in the A83-01 + LPS treatment group, as A83-01 significantly reduced the viability of LPS-induced septic mice. ASCs had no effect on improving survival in the LPS-treated group and only improved survival in the A83-01 + LPS-treated group. Taken together, our results suggest that the co-administration of A83-01 and ASCs in vivo is not advisable, and further studies are needed to evaluate whether secretomes from ASCs exposed to 4F + A83-01 in vitro can be used to alleviate sepsis.

Furthermore, we found that 4F and A83-01 significantly increased the expression of TSG-6 in ASCs, but in ASCs co-cultured with macrophages, A83-01 did not increase the expression of TSG-6. Therefore, it is important to understand why TSG-6 expression was differentially regulated in the two experiments in order to develop TSG-6 as a therapeutic option in ASCs for inflammatory diseases. Indeed, in experiments determining the effects of anti-inflammatory TGF-β, IL-10, MSC-produced IL-6, or prostaglandin E2 (PGE2) on the expression of TSG-6 in ASCs, it was observed that TGF-β decreased 4F-induced TSG-6 expression, whereas PGE2 increased 4F-induced TSG-6 expression (Figure 6A). Furthermore, celecoxib, a Cox-2 inhibitor, decreased 4F-induced TSG-6 expression (Figure 6B). Therefore, when ASCs were co-cultured with macrophages, it was predicted that A83-01 would not induce an increase in TSG-6 expression by decreasing the expression of Cox-2, which in turn decreases the production of PGE2. Additionally, kynurenic acid, an IDO metabolite, can regulate the expression of TSG-6 in MSCs [75]. SB202190 decreased the expression of IDO2 and increased the expression of 4F-induced TSG-6 (Figure 5). The IDO inhibitor 1-methyl tryptophan also increased the 4F-induced TSG-6 expression (Figure 6B). These results demonstrate that the IDO metabolites act as a negative regulator of TSG-6 expression in MSCs. However, it should be noted that SB202190 decreased Cox-2 expression, leading to a decrease in the 4F-induced TSG-6 expression. To summarize, the combination of A83-01 or SB202190 in ASC transplantation may be undesirable because it may reduce the expression of Cox-2 and IDO2 in ASCs, thereby reducing the therapeutic effect of ASCs. Therefore, whether the appropriate combination of A83-01, SB202190, PGE2, and 1-methyl tryptophan can be used in ASC transplantation therapy or whether ASCs overexpressing TSG-6 or their secretion can be used to control sepsis is an issue warranting further investigation.

These results suggest that the therapeutic effect of MSCs may vary depending on the intensity of inflammation, suggesting that the timing of MSC transplantation and the number of transplanted MSCs should be determined comprehensively by evaluating the expression of immunomodulatory factors including TNF-α, IL-1β, and TGF-β. Furthermore, when evaluating the therapeutic effect of stem cells on inflammation intensity ex vivo, it is preferable to use at least co-culture conditions with macrophages or other inflammatory cells.

## 4. Materials and Methods

### 4.1. Reagents

The reagents used in this study were obtained from the indicated suppliers: IFN-γ, TNF-α, IL-1β, TGF-β, and TSG-6 from R&D Systems (Minneapolis, MN, USA); antibodies against GAPDH (sc47724), TSG-6 (sc377277), Cox-2 (sc1747), and IDO2 (sc87164) from Santa Cruz Biotechnology (Santa Cruz, CA, USA); antibodies against pp38 (4511S), SMAD2/3 (8685S), and cIL-1β (83186S) from Cell Signaling Technology (Danvers, MA, USA); antibodies against pSmad2/3 (PA5-110155) from Thermo Fisher Scientific (Waltham, MA, USA); and chemical inhibitors for JNK (SP600125) from MedChemExpress, Princeton, NJ, USA; p38 (SB202109) from MedChemExpress; ERK (PD98059), ROCK (Y-27632), and TGF-β type I receptor (A83-01) from Sigma-Aldrich (St. Louis, MO, USA) for cell culture or MedChemExpress for animal study; and NF-κB (Bay 11-7082), GSK3β (CHIR99021), and PI3K (LY294002) from Sigma-Aldrich. All other materials were purchased from Sigma-Aldrich.

### 4.2. Cell Culture

THP-1 monocytes were purchased from the Korean Cell Line Bank (Seoul, Republic of Korea) and maintained in Roswell Park Memorial Institute 1640 Medium (Gibco BRL/Thermo Fisher Scientific, Waltham, MA, USA) supplemented with 10% fetal bovine serum (FBS) and penicillin/streptomycin (Gibco BRL/Thermo Fisher Scientific). For differentiation into macrophages, THP-1 cells were treated with 100 nM of phorbol 12-myristate-13-acetate for 2 days, and for transition into M1 macrophages, they were treated with 20 ng/mL of IFN-γ and 10 pg/mL of LPS for an additional 2 days. ASCs were isolated from three healthy donors (24–38 years of age) with their written informed consent through elective liposuction procedures under anesthesia at the Wonju Severance Christian Hospital (Wonju, Republic of Korea). ASCs were isolated using a modified protocol as described by Zuk et al. [76] and subcultured in Dulbecco’s Modified Eagle Medium (Gibco BRL/Thermo Fisher Scientific) with 10% FBS and penicillin/streptomycin; ASCs at less than five passages were used in this study.

For the indirect co-culture of ASCs and macrophages, ASCs were seeded in 6-well plates 1 day before, macrophages were allowed to undergo differentiation in the upper chamber of Transwell plates (SPL Life Sciences, Pocheon, Republic of Korea) 2 days before, and then the upper chamber was mounted in 6-well plates seeded with ASCs. Unless otherwise indicated in the figures, the following concentrations of reagents were used: 20 ng/mL of IFN-γ, IL-10, and IL-6; 10 pg/mL of LPS; 1 ng/mL of TNF-α, IL-1β, and TGF-β; 80 ng/mL of TSG-6; 1 μM of SP600125, SB202109, Y-27632, and CHIR99021; 2 μM of PD98059; 500 nM of A83-01, 0.2 μM of Bay 11-7082, LY294002, and PGE2, 0.1 mM of 1-methyl tryptophan, and 20 μM of celecoxib.

### 4.3. Animal Study

All animal experiments were performed in accordance with the institutional guidelines and approved by the Institutional Animal Care and Use Committee of the Yonsei University Mirae Campus, Wonju (YWCI-202307-011-01). Six-week-old C57BL/6 mice were purchased from RaonBio (Yongin, Republic of Korea). Mice were housed in ventilated cages on a 12 h light/dark cycle and were acclimatized for 1 week before experimentation. LPS from *Escherichia coli* O111:B4 (L4130, Sigma-Aldrich) was dissolved in phosphate-buffered saline and 200 μL (12 mg/kg) was injected into the peritoneal cavity to induce septic shock. Mice were divided into the following groups and different numbers of mice were assigned to each group according to their expected survival: sham control group, *n* = 5; LPS group, *n* = 15; LPS + A83-01 group, *n* = 14; LPS + ASC group, *n* = 10; and LPS + A83-1 + ASC group, *n* = 10. Three hours after LPS injection, A83-01 (0.5 mg/mouse) and/or ASCs (1 × 10^6^ cells/mouse) were administered intraperitoneally in that order. The survival rate of each group was measured until day 3, at which time the surviving mice were euthanized by CO_2_ asphyxiation and peritoneal cells and serum were collected.

### 4.4. Immunoblotting

The total protein of ASCs or macrophages was extracted using sample buffer [62.5 mM Tris-HCl, pH 6.8, 34.7 mM sodium dodecyl sulfate (SDS), 5% β-mercaptoethanol, and 10% glycerol], and equal volumes of the protein samples were separated by 10% or 12% SDS-polyacrylamide gel electrophoresis and transferred to polyvinylidene difluoride membranes (Millipore, Burlington, MA, USA). The membrane was blocked with 5% skim milk or 5% bovine serum albumin in Tris-HCl-buffered saline containing 0.05% Tween 20 (TBST) for 30 min, and incubated with primary antibodies against TSG-6, Cox-2, IDO2, and GAPDH (Santa Cruz Biotechnology, 1:1000); pp38, SMAD2/3, and cIL-1β (Cell Signaling Technology, 1:1000); and pSmad2/3 (Thermo Fisher Scientific) at 4 °C overnight, followed by incubation with horseradish peroxidase-conjugated secondary anti-rabbit or anti-mouse antibodies (Cell Signaling Technology, 1:5000) for 1 h at room temperature. After primary and secondary antibody incubation, the membranes were washed thrice for 5 min with TBST and then treated with an EZ-Western Lumi Pico or EZ-Western Lumi Femto (DoGenBio, Seoul, Republic of Korea). The target bands were detected using a ChemiDoc XRS+ System (Bio-Rad Laboratories, Hercules, CA, USA).

### 4.5. Quantitative PCR

Total RNA from cells was extracted using the AccuPrep Universal RNA Extraction Kit (Bioneer, Daejeon, Republic of Korea) and cDNA was synthesized from 1 μg of total RNA using the Verso cDNA Synthesis Kit (Thermo Fisher Scientific) according to the manufacturer’s instructions. The cDNA was mixed with primer mix and Power SYBR Green PCR Master Mix (Applied Biosystems, Dublin, Ireland) and amplified using the QuantStudio 6 Flex Real-time PCR system (Thermo Fisher Scientific). The 2^−(△△Ct)^ method was used to measure the relative fold changes in mRNA expression. The sequences of sense and antisense primers were as follows: 5′-CAGGCTGCTCTGGGATTCTC-3′ and 5′-GTCCTGGAAGGAGCACTTCAT-3′ for IL-1β, 5′-AAGTGGACATCAACGGGTTC-3′ and 5′-GTCCAGGCTCCAAATGTAGG-3′ for TGF-β, and 5′-CAAGGCTGAGAACGGGAAGC-3′ and 5′-AGGGGGCAGAGATGATGACC-3′ for GAPDH.

### 4.6. Flow Cytometry

Macrophages were stained with phycoerythrin-conjugated antibodies against CD68 or CD206 (BD Biosciences, San Jose, CA, USA) in the dark for 20 min at room temperature. Phycoerythrin-conjugated mouse immunoglobulin G was used as the control isotype at the same concentration. The fluorescence intensity of the cells was evaluated by flow cytometry (FACS Aria III Cell Sorter, BD Biosciences, Franklin Lakes, NJ, USA) and the data were analyzed using the FACSDiva Software v8.5 (BD Biosciences).

### 4.7. Statistical Analyses

All statistical analyses were performed using one-way analysis of variance, followed by Tukey’s post hoc tests. A two-tailed Student’s *t*-test was used to evaluate the differences between two groups. Data are presented as the mean ± SD. All *p* values less than 0.05 were considered statistically significant.

## 5. Conclusions

TGF-β decreased TSG-6 expression in ASCs, but TGF-β type 1 receptor inhibitor A83-01 and p38 MAPK inhibitor SB202190 could significantly increase TSG-6 expression in ASCs. However, A83-01 further exacerbated LPS-induced sepsis in vivo. ASCs significantly increased the survival rate of LPS-induced septic mice whose sepsis was exacerbated by A83-01, but did not increase it beyond that of LPS-induced septic mice. These results can be attributed to A83-01 and SB202190 inducing reduced expression of Cox-2 and IDO2 in ASCs co-cultured with macrophages. Therefore, when ASCs are used to treat inflammatory diseases, co-administration of A83-01 and SB202190, which may affect the inflammatory response, is not recommended.

## Figures and Tables

**Figure 1 ijms-25-00477-f001:**
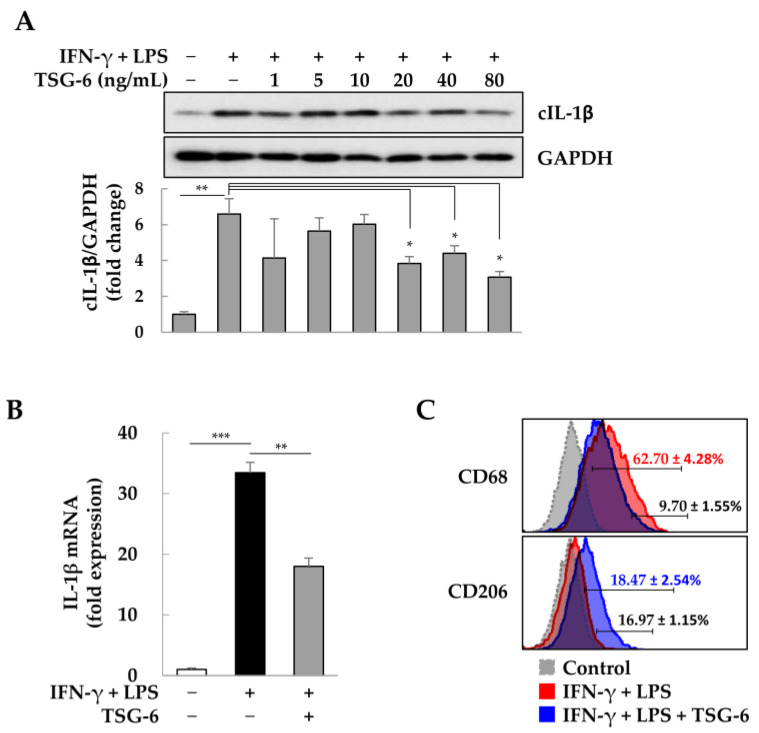
Anti-inflammatory effects of tumor necrosis factor-α (TNF-α)-stimulated gene 6 (TSG-6) in M1 macrophages. Phorbol 12-myristate-13-acetate (PMA)-treated THP-1 cells (macrophages) were further treated with interferon-γ (IFN-γ) + lipopolysaccharide (LPS) in the presence or absence of TSG-6 (0–80 ng/mL) for 2 days. (**A**) Expression of cleaved interleukin-1β (cIL-1β) induced by TSG-6. The intensity of protein expression was quantified through densitometry in ImageJ (National Institutes of Health, https://imagej.net/ij/index.html, 10 August 2023), and its relative expression was normalized against that of glyceraldehyde 3-phosphate dehydrogenase (GAPDH). Data are presented as the mean ± standard deviation (SD) of three independent experiments. (**B**) Expression of *IL-1β* mRNA induced by TSG-6. *IL-1β* expression was normalized using *GAPDH* expression. Relative fold changes in mRNA expression were measured using the 2^−(△△Ct)^ method. The results are presented as the mean ± SD of three replicates. (**C**) M1-to-M2 transition of macrophages induced by TSG-6. Data are presented as the mean ± SD of three independent experiments. * *p* < 0.05, ** *p* < 0.01, and *** *p* < 0.001.

**Figure 2 ijms-25-00477-f002:**
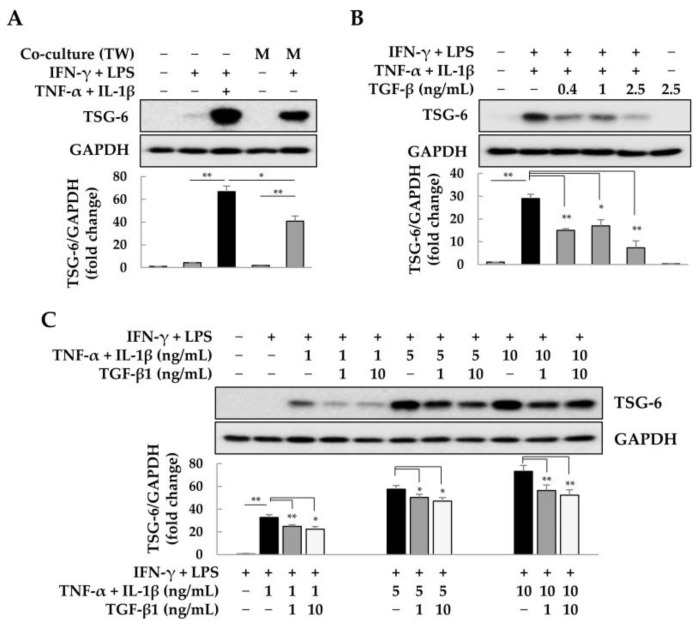
Suppression of tumor necrosis factor-α (TNF)-α stimulated gene-6 (TSG-6) expression by transforming growth factor-β (TGF-β) in adipose tissue-derived stem cells (ASCs). ASCs were indirectly co-cultured with M1 macrophages or exposed to pro-inflammatory factors (interferon [IFN]-γ, lipopolysaccharide [LPS], TNF-α, or interleukin [IL]-1β) for 2 days. TSG-6 expression was analyzed by immunoblotting and expression changes were analyzed using ImageJ (National Institutes of Health, https://imagej.net/ij/index.html, 10 August 2023). (**A**) TSG-6 expression in ASCs in response to pro-inflammatory factors or M1 macrophages. (**B**) Decrease in TSG-6 expression in ASCs by TGF-β. (**C**) Expression of TSG-6 in ASCs in response to the concentration of pro-inflammatory factors and TGF-β. ASCs were treated with IFN-γ (20 ng/mL) and LPS (10 pg/mL) and further treated with TNF-α and IL-1β (each 1, 5, or 10 ng/mL) and/or TGF-β (1 or 10 ng/mL) for 2 days. Data are presented as the mean ± SD of three independent experiments. The intensity of protein expression was quantified through densitometry in ImageJ, and its relative expression was normalized against that of GAPDH. * *p* < 0.05 and ** *p* < 0.01. TW, Transwell; M, macrophages.

**Figure 3 ijms-25-00477-f003:**
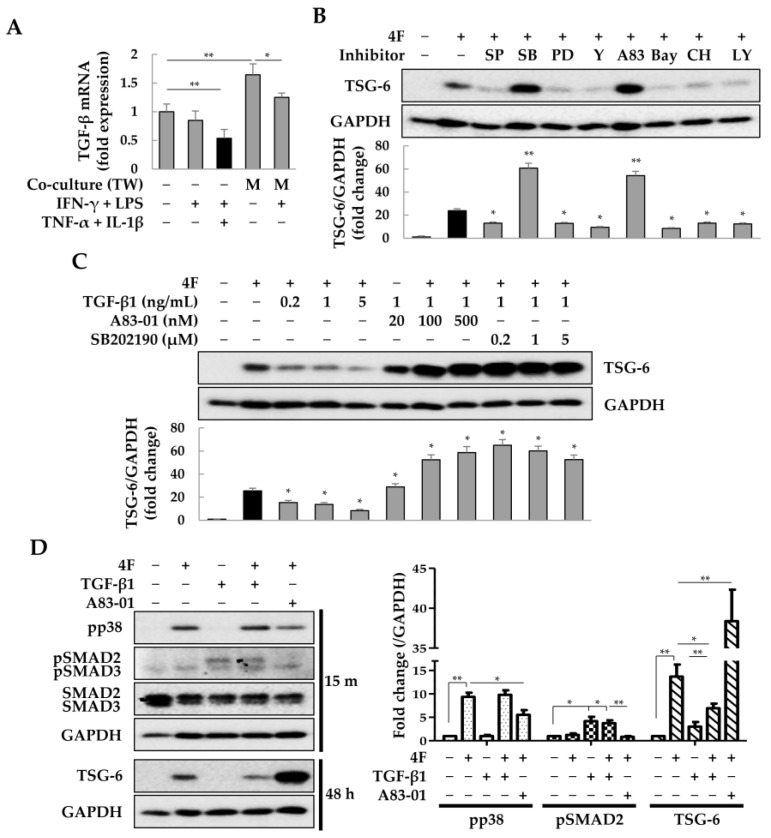
Effects of A83-01 and SB202190 on tumor necrosis factor-α (TNF)-α stimulated gene-6 (TSG-6) expression in adipose tissue-derived stem cells (ASCs). ASCs were treated with four pro-inflammatory factors (4F), IFN-γ, LPS, TNF-α, or IL-1β, or co-cultured with macrophages and the mRNA expression of TGF-β was determined using quantitative PCR. The TSG-6 protein was analyzed by immunoblotting after treatment of ASCs with inhibitors of proteins involved in the TGF-β signal pathway for 2 days. (**A**) *TGF-β* mRNA in ASCs treated with pro-inflammatory factors or co-cultured with macrophages. Target gene expression was normalized using *GAPDH* expression. Relative fold changes in *TGF-β* mRNA were measured using the 2^−(△△Ct)^ method. The results are presented as the mean ± SD of three replicates. * *p* < 0.05 and ** *p* < 0.01. (**B**) TSG-6 expression in ASCs treated with 4F but without TGF-β. (**C**) TSG-6 expression in ASCs treated with 4F and A83-01 or SB202190 after addition of TGF-β. (**D**) Phosphorylation of p38 (pp38) and SMAD2 (pSMAD2) by 4F and TGF-β. Data in (**B**–**D**) are presented as the mean ± SD of three independent experiments. * *p* < 0.05 and ** *p* < 0.01 vs. 4F. TW, Transwell; M, macrophages; SP, SP600125; SB, SB202190; PD, PD98059; Y, Y-27632; A83, A83-01; Bay, Bay 11-7082; CH, CHIR99021; LY, LY294002.

**Figure 4 ijms-25-00477-f004:**
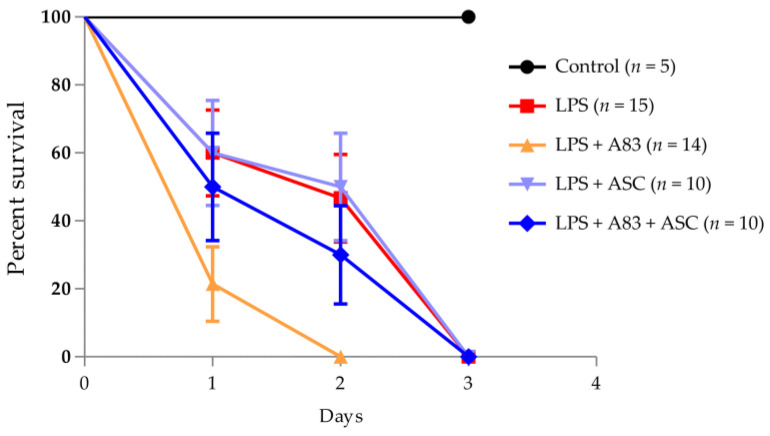
Survival rate of LPS-induced septic mice with A83-01 treatment and/or adipose tissue-derived stem cells (ASC) transplantation. The sepsis model was created by intraperitoneal injection of LPS (15 mg/kg). Three hours later, A83-01 (0.5 mg/mouse) and/or ASCs (1 × 10^6^ cells/mouse) were administered into the peritoneal cavity in that order.

**Figure 5 ijms-25-00477-f005:**
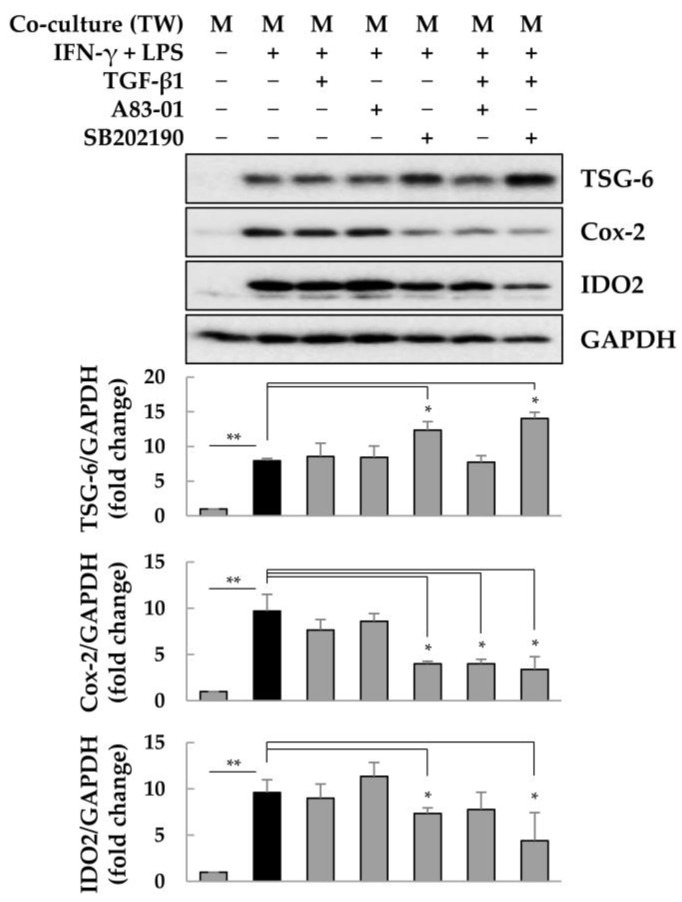
Expression of TSG-6, Cox-2, and IDO2 in ASCs co-cultured with macrophages. ASCs were co-cultured with M1 macrophages for 2 days in the presence or absence of TGF-β1, A83-01, or SB202190. The intensity of protein expression was quantified through densitometry in ImageJ (National Institutes of Health, https://imagej.net/ij/index.html, 10 August 2023), and its relative expression was normalized against that of GAPDH. Data presented as the mean ± SD of three independent experiments. * *p* < 0.05 and ** *p* < 0.01. TW, Transwell; M, macrophages.

**Figure 6 ijms-25-00477-f006:**
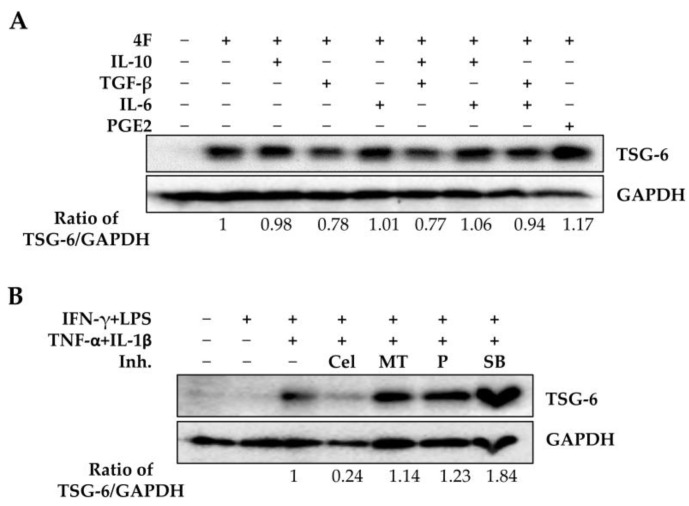
Effects of IDO- or Cox-2-related metabolites on TSG-6 expression in ASCs. ASCs were treated in the presence of four pro-inflammatory factors (4F): IFN-γ, LPS, TNF-α, or IL-1β, with and without the indicated inhibitors or cytokines for two days. TSG-6 protein levels were analyzed by immunoblotting. (**A**) TSG-6 expression in 4F-treated ASCs with TGF-β or PGE2. (**B**) TSG-6 inhibitors of Cox-2 (Cel), IDO (MT), or p38 MAPK (SB). Cel, Celecoxib; MT, 1-methyl tryptophan; P, PGE2; SB, SB202190.

## Data Availability

Data is contained within the article.

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
