# Peer review of "Role of TGF-β and p38 MAPK in TSG-6 Expression in Adipose Tissue-Derived Stem Cells In Vitro and In Vivo"

_ijms, 2023, doi:10.3390/ijms25010477_

Round 1

Reviewer 1 Report

Comments and Suggestions for Authors

This work investigates the modulation of inflammation via signaling interactions between macrophages differentiated from THP-1 monocytes and mesenchymal stromal cells isolated from adipose tissue (ASCs), then describes a suppressive effect of TGF-ß on the expression of TSG-6 by ASCs. Based on this, mouse experiments were carried out to see whether coapplication of ASCs and the TGF-ßR1 inhibitor A83-01 can reduce damage in an LPS-induced acute inflammation in mice.

The methodology used for teh in vitro work is largely sound and the results are clearly reported.

The introduction should describe what is known concerning the antiinflammatory potential of adipose versus bone marrow MSCs and explain why mesenchymal stromal cells from adipose tissue were chosen, rather than those from bone marrow.

The reasoning that is used to justify the focus on the TGFß – TSG-6 axis as the decisive regulator of the MSC response to inflammation is weak. Given that TGF-ß has a plethora of both pro-and anti-inflammatory effects on a wide range of cell types; that the MSC response to inflammation is complex and mutlifactorial and that feedback regulation of inflammation is likely to comprise a set of interacting checks and balances that change over time, the hope that coapplication of a TGF-ß inhibitor with ASC will rescue mice from acute inflammation was very optimistic.

The mouse experiment itself also has some important limitations. Firstly, the MSC alone had no effect whatsoever in the model used. Strictly speaking, one cannot conclude whether or not TGF-ß inhibition has the potential to enhance an MSC function if that function is absent in the „MSC only“ control condition.  There should have been dose finding experiments to establish a model in which MSC have at least some ability to limit an inflammatory response.

Secondly, simultaneous coapplication is just one of the possible treatment regimens. In this setting, it may have been more relevant to expose the MSC to TGF-ß in vitro, wash, and then infuse the cells, thus avoiding the comlications caused by a barrage of TGF-ß effects on other cell types.

Reviewer 2 Report

Comments and Suggestions for Authors

The authors present a relevant study on how TGFb modulation can alter the expression of TGS-6 in ASC and influence the immunological response, aiming to improve the effectiveness of MSC therapy. The study is well presented, but some points could be clarified.

1.       In Figure 1, was only IL1b affected by TSG-6? It would be interesting to include information about the expression of other pro-inflammatory cytokines and also a graph with the total % of cells stained with CD68 and CD206.

2.      Line 130 – Please check if the Figure citation is correct.

3.      Figure 3B and lines 159-161 - it is not clear why all these inhibitors were tested. Please clarify.

4.      In lines 169-172, it is indicated that the combined effect of A83-01 and SB202190 could be interesting. Have in vitro assays been performed showing the effect of the combination of these inhibitors?                        

5.      Line 212 – The authors state that the A83-01 reduced the expression of Cox 2 and TSG-6 (Figure 5 Lane 6). However, the quantification shows no difference in TSG-6 expression between Lane 6 and the condition only with TGFb1 or A83-01 (Lanes 3 and 4). Please review this statement.

6.      The results using the 4 factors or co-culture with macrophages with ASCs were different. The authors could discuss more about this. Is the concentration of factors used similar to that released by M1 macrophage? Could there be other cytokines released that interfere with the ASC response?

7.      Some experiments were performed only with A83-01. Furthermore, the relationship between SB202190 and TGF and the context of inflammation are not very clear in the text.

8.      To explain the in vivo results, it is shown that SB202190 reduced the expression of COX-2 and IDO2 in ASC co-culture with macrophages. However, the in vivo experiments were performed only with A83-01. So, it is not possible to state that Cox-2 and IDO were reduced in this case.

9.      Considering that in the introduction it was highlighted that the study aimed to identify optimal conditions for the expression of TGS-6 in ASCs, why not use ASC pre-treated with the inhibitors (and improved in TSG-6 expression) in in vivo assays? Couldn't systemic administration of the inhibitor affect other cells differently? Would it be possible to use the ASC secretome?

10.   The discussion could be improved, for example, by including other variables that may have influenced the results, adding comparisons with other studies that corroborate, or not, the results obtained. TGFb secreted by MSC has a role in reducing inflammation. Could reducing its activation in MSC change the cell's characteristics? Why do the authors believe that ASCs did not improve the survival rate in the sepsis model (it is the in vivo model, number of cells)?

11.   What is the ASC isolation protocol? It would be interesting to cite the protocol or a reference study.

12.   The study has many variables and conditions. It would be interesting to include a figure or graphical abstract summarizing the results.

Round 2

Reviewer 1 Report

Comments and Suggestions for Authors

The authors have now provided acceptable responses to the issues that I raised 

Author Response

We would like to thank the reviewer once again for his/her insightful and constructive critique of our work.

Reviewer 2 Report

Comments and Suggestions for Authors

Thank you for reviewing your manuscript. There are still some points that need to be clarified.

1. The authors indicated in the response (topics 5 and 8) that they removed TSG-6 (now line 234) and IDO2 (line 231) from the text, however in the version received they were still maintained. Please check again.

2. The results included are interesting but are in the Discussion section (lines 287-307). Please include them in the Results section. Furthermore, explain a little more why using IL10, IL6, and PGE2 and how it relates to the difference in response between macrophages and 4F.

3. I believe the manuscript Discussion section would greatly benefit from some of the discussions included in the response, for example in topic 10. Furthermore, the authors indicated several possibilities for future studies in the response. It would be interesting to include these perspectives in the manuscript.

Round 3

Reviewer 2 Report

Comments and Suggestions for Authors

Thank you for reviewing your manuscript.